# RGFT: Broadening Correct Reasoning Paths via Reference-Guided Fine-Tuning

## Abstract

Rollout-based reinforcement learning has become a dominant approach for improving mathematical reasoning in large language models, but its reliance on large-scale sampling and self-play makes it expensive and difficult to deploy. Recent rollout-free alternatives instead fine-tune models using only verified correct solutions, yet often collapse effective coverage over valid reasoning paths under positive-only supervision. In this work, we study how to broaden correct reasoning paths in a strictly offline setting without rollouts or negative feedback. We identify the absence of an explicit probability scale over correct solutions as a key limitation of existing rollout-free objectives. We propose Reference-Guided Fine-Tuning (RGFT), a rollout-free framework that introduces a learnable reference policy to define a relative probability scale over verified reasoning trajectories. A ratio-guided objective anchored to this reference encourages the target policy to upweight correct but underrepresented reasoning paths, while a stochastic reference–target mixing scheme prevents collapse under shared initialization. Across multiple mathematical reasoning benchmarks and model scales, RGFT consistently outperforms supervised fine-tuning and recent rollout-free reweighting methods. In particular, RGFT achieves stronger performance under sampling-based evaluation on tasks admitting multiple valid reasoning paths, while maintaining substantially lower distributional drift from the base model.

## 1. Introduction

Recent advances in large language models have shown that strong mathematical reasoning often relies on the ability to explore and maintain multiple valid solution paths rather than committing prematurely to a single dominant trajectory. For many reasoning problems, especially those with structured or symbolic solutions, there exist diverse chains of reasoning that are all correct but differ in intermediate steps, abstraction level, or decomposition strategy. An effective post-training procedure should therefore improve correctness while preserving sufficient diversity in the learned policy, avoiding overly sharp distributions that hinder generalization and robustness.

In practice, post-training of reasoning models is commonly performed using a combination of supervised fine-tuning and reinforcement learning (Wei et al., 2021; Ouyang et al., 2022; Chung et al., 2024). While reinforcement learning with explicit rewards can encourage exploration, it typically relies on large-scale rollouts or self-play, which are expensive and often impractical at scale (Christiano et al., 2017; Schulman et al., 2015; Schulman et al., 2017). This has motivated a growing body of work on rollout-free or offline alternatives (Levine et al., 2020), particularly in settings where a verifier is available and training data consists exclusively of correct solutions. In such positive-only regimes, however, the learning signal is inherently limited: all training examples are equally correct, and there is no direct feedback indicating which correct solutions are underrepresented, fragile, or hard to learn.

Several recent methods address this limitation by reweighting supervised objectives to better approximate sparse-reward reinforcement learning (Qin & Springenberg, 2025; Wu et al., 2025). Self-weighted approaches adjust gradient magnitudes based on the model's own probabilities, while trust-region-inspired methods stabilize training by constraining updates relative to a fixed reference policy (Zhu et al., 2025; Zhu et al., 2025). Although effective in many cases, these approaches face a common tension. Self-weighting tightly couples the optimization signal to the current model state and may amplify biases or collapse support, while fixed-reference methods rely on an externally given policy that serves only as a regularizer and does not capture the distribution over *correct reasoning paths*. In particular, the choice of a reference distribution over correct reasoning paths is typically implicit or assumed, rather than explicitly modeled.

[1]Anonymous Institution, Anonymous City, Anonymous Region, Anonymous Country. Correspondence to: Anonymous Author <anon.email@domain.com>.

Preliminary work. Under review by the International Conference on Machine Learning (ICML). Do not distribute.

In this work, we focus on broadening correct reasoning paths in rollout-free fine-tuning by addressing the role of the reference policy directly. Instead of treating the reference as fixed or implicit, we propose **Reference-Guided Fine-Tuning (RGFT)**, which introduces a learnable reference policy that is jointly optimized alongside the target policy using strictly positive demonstrations. The reference policy is not assumed to be optimal; rather, it serves as a probability scale anchored on verified solutions. Guided by this scale, the target policy is encouraged to increase the probability of correct reasoning paths that are underrepresented by the reference, promoting broader coverage of the solution space without relying on self-weighting or external rollouts.

A practical challenge in this joint optimization setting is that, under shared initialization, the target and reference policies can evolve in a highly correlated manner, effectively reducing the procedure to standard supervised fine-tuning. To mitigate this effect, we introduce a simple optimization-level technique: a randomized mixture of the reference and target objectives using per-token coefficients sampled from a Beta distribution. This stochastic mixing does not alter the expected training objective, but it significantly improves optimization dynamics by reducing gradient alignment between the two policies, enabling them to assume distinct functional roles during training.

Finally, our approach differs from many reinforcement-learning-based post-training methods in its effect on policy entropy(Yue et al., 2025; Gao et al., 2022). Rather than aggressively concentrating probability mass on a narrow set of preferred trajectories, the combination of reference anchoring and ratio-guided updates avoids excessive entropy reduction. As a result, the fine-tuned models retain more of the base model's generalization capability while exhibiting improved reasoning behavior. Empirical results across a range of mathematical benchmarks and model scales demonstrate that this design leads to consistent gains over existing reweighting-based fine-tuning methods, without the need for rollouts or additional supervision.

In summary, our contributions are as follows:

- We introduce RGFT, a reference-guided offline fine-tuning framework that explicitly learns a reference policy from strictly positive demonstrations, improving upon existing rollout-free reweighting methods that rely on fixed or implicit references.

- We propose a ratio-guided objective that leverages the learned reference as a probability scale, encouraging broader coverage of correct reasoning paths without self-weighted coupling or explicit rollouts.

- We present a simple yet effective optimization-level technique based on Beta-distributed stochastic mixing

to stabilize joint learning and prevent collapse under shared initialization.

- Our approach avoids excessive entropy reduction commonly observed in reinforcement-learning-style post-training, thereby preserving more of the base model's generalization ability while improving reasoning performance.

## 2. Related Work

Large language models are typically trained through a combination of supervised fine-tuning (SFT) and reinforcement learning (RL) based post-training. Instruction tuning and SFT align pretrained models to follow human instructions by fitting curated input–output pairs, forming the foundation of most modern LLM pipelines (Wei et al., 2021; Ouyang et al., 2022; Chung et al., 2024). To further improve alignment, robustness, and reasoning performance, SFT is often followed by RL-based post-training, most notably reinforcement learning from human feedback (RLHF), where policy optimization methods such as TRPO and PPO are used to maximize preference-based or verifier-defined rewards (Christiano et al., 2017; Ouyang et al., 2022; Schulman et al., 2015; Schulman et al., 2017). More recent variants, including direct preference optimization (DPO), group relative policy optimization (GRPO), and verifier-based RL for reasoning tasks, retain the core idea of optimizing a policy with respect to an explicit reward signal, but still rely on policy rollouts or large-scale sampling (Cobbe et al., 2021; Lewkowycz et al., 2022; Lightman et al., 2023; Rafailov et al., 2023; Qwen Team, 2024; Guo et al., 2025).

Despite their effectiveness, rollout-based RL methods are computationally expensive and often prohibitively costly for large language models, motivating increasing interest in rollout-free or offline alternatives. This line of work draws heavily on ideas from off-policy and offline reinforcement learning, where policies are optimized using static datasets collected from a different behavior policy. Classic off-policy methods rely on importance sampling to correct distribution mismatch between the data-generating policy and the target policy, while trust region methods constrain policy updates to ensure stability (Precup et al., 2000; Sutton et al., 1998; Levine et al., 2020; Schulman et al., 2015; Schulman et al., 2017). These principles provide a natural conceptual foundation for offline fine-tuning of language models, especially in positive-only regimes where the goal is to improve correctness while maintaining broad coverage over correct solutions.

Building on this perspective, several recent works reinterpret supervised fine-tuning on positive demonstrations as optimizing a surrogate objective for sparse-reward RL. Importance-weighted supervised fine-tuning (iw-SFT) for-

malizes SFT as optimizing a lower bound of a sparse-reward RL objective and proposes importance reweighting to tighten this bound under distribution shift (Qin & Springenberg, 2025). Dynamic fine-tuning (DFT) further explores probability-based reweighting schemes that modify the gradient structure of SFT to better align with policy-gradient updates (Wu et al., 2025). These approaches demonstrate that carefully designed reweighting can significantly improve performance without explicit rollouts. However, self-weighted objectives inherently couple the optimization signal to the current model state, which may amplify biases and lead to collapse of effective coverage over correct reasoning paths.

Another line of work focuses on stabilizing reweighted objectives through reference-based or trust-region-inspired constraints. Anchored supervised fine-tuning (ASFT) introduces explicit KL regularization to a fixed base policy in order to control distributional drift in self-weighted objectives (Zhu et al., 2025). Proximal supervised fine-tuning (PSFT) further emphasizes the role of old or reference policies and shows that the frequency and manner in which the reference is updated can substantially affect optimization stability (Zhu et al., 2025). While effective, these methods typically treat the reference policy as fixed or externally given, using it primarily for stabilization rather than to represent and shape the distribution over correct solutions.

Overall, existing rollout-free fine-tuning methods demonstrate that off-policy and trust-region ideas can approximate the benefits of RL post-training using static positive data alone. At the same time, they leave open the question of how a reference distribution over correct solutions should be defined, maintained, or learned when only verified positive demonstrations are available. In particular, the reference over correct reasoning paths is typically implicit in the dataset construction process and cannot be assumed optimal a priori, yet it critically affects which correct solutions the model covers after fine-tuning. These limitations motivate reference-guided approaches that go beyond fixed or implicit references and instead address the role of the reference policy directly within a single-stage offline training framework.

## 3. Method

We propose **Reference-Guided Fine-Tuning (RGFT)**, a single-stage offline fine-tuning framework for *broadening correct reasoning paths* under positive-only supervision. RGFT jointly learns a target policy and a learnable reference policy from strictly positive demonstrations, using a randomized mixture objective to prevent collapse under shared initialization.

### 3.1. Problem Setting and Motivation

We consider an offline learning setting where the training dataset $\mathcal{D} = \{(x, y)\}$ consists exclusively of *positive* examples, i.e., each output $y$ is a verified correct solution for the corresponding input $x$. No rollouts, self-play, or reward models are used during training. Our goal is to learn a target policy $\pi_\theta(y \mid x)$ that produces correct solutions *while maintaining broad coverage over correct reasoning paths* under this strictly positive supervision regime.

A fundamental difficulty in this setting is that positive-only data provides no explicit signal for relative preference among correct solutions. Standard supervised fine-tuning (SFT) therefore reduces to fitting the empirical distribution of demonstrated solutions, without offering a mechanism to emphasize solutions that are correct but underrepresented or harder to model. Recent approaches attempt to address this limitation by reweighting gradients using the model's own probabilities or by constraining updates relative to a fixed reference. However, these approaches either suffer from self-coupling effects or rely on a reference policy whose role is fixed and whose induced distribution over correct solutions is implicit.

This motivates the introduction of an explicit reference policy $\pi_{\text{ref}}$ that provides a relative scale over *correct reasoning paths*, while avoiding assumptions that this reference is optimal or externally given.

### 3.2. From Reward Maximization to Ratio Guidance

We start from the natural objective of maximizing the success probability of the target policy on verified solutions. Let $r(x, y) \in \{0, 1\}$ denote a verifier-defined reward indicating whether $y$ is correct for $x$. The ideal objective is

$$J(\theta) = \mathbb{E}_x \mathbb{E}_{y \sim \pi_\theta(\cdot \mid x)} \big[ r(x, y) \big]. \tag{1}$$

In our setting, $\mathcal{D}$ contains only verified solutions, but we do not have access to rollouts $y \sim \pi_\theta$ nor to any incorrect samples. To relate (1) to an offline objective, we introduce an explicit reference policy $\pi_{\text{ref}}$ and rewrite the expectation using importance weighting:

$$J(\theta) = \mathbb{E}_x \mathbb{E}_{y \sim \pi_{\text{ref}}(\cdot \mid x)} \left[ r(x, y) \frac{\pi_\theta(y \mid x)}{\pi_{\text{ref}}(y \mid x)} \right]. \tag{2}$$

Here $\pi_{\text{ref}}$ is understood as a probabilistic description of the data distribution $\mathcal{D}$, namely the conditional distribution over verified solutions observed in the dataset. Accordingly, we only require $\pi_{\text{ref}}(y \mid x)$ to be well-defined on the support of $\mathcal{D}$, where it serves as a relative probability scale for the ratio-guided update. In later sections, we describe how such a reference distribution can be instantiated and learned in practice within a single-stage training framework.

Since $\mathcal{D}$ contains only verified solutions, we treat each $(x, y) \in \mathcal{D}$ as having $r(x, y) = 1$ and absorb the reward into the offline surrogate objective. This leads to the reference-guided ratio term

$$\mathcal{L}_{\text{ratio}} = -\mathbb{E}_{(x,y)\sim\mathcal{D}}\left[\frac{\pi_\theta(y \mid x)}{\text{sg}(\pi_{\text{ref}}(y \mid x))}\right], \qquad (3)$$

where $\text{sg}(\cdot)$ denotes the stop-gradient operator. The reference policy appears only as a fixed normalization factor in the denominator and does not receive gradients from $\mathcal{L}_{\text{ratio}}$.

The ratio-guided objective in (3) recovers several existing rollout-free objectives under different choices of the denominator. Standard supervised fine-tuning (SFT) minimizes

$$\mathcal{L}_{\text{SFT}} = -\mathbb{E}_{(x,y)\sim\mathcal{D}}[\log \pi_\theta(y \mid x)].$$

Dynamic Fine-Tuning (DFT) adopts a self-weighted form

$$\mathcal{L}_{\text{DFT}} = -\mathbb{E}_{(x,y)\sim\mathcal{D}}[\text{sg}(\pi_\theta(y \mid x)) \log \pi_\theta(y \mid x)].$$

Within (3), setting $\text{sg}(\pi_{\text{ref}}) \equiv 1$ yields a gradient equivalent to DFT, while choosing the denominator as $\text{sg}(\pi_\theta)$ produces a gradient equivalent to SFT. These equivalences hold at the level of gradients under the stop-gradient operator, rather than as pointwise numerical identities. In both cases, the induced reference scale is either fixed or self-coupled and does not represent an explicit, data-grounded reference distribution over verified solutions.

### 3.3. Learnable Reference Policy

Existing rollout-free methods typically rely on either implicit or fixed reference mechanisms. Implicit references, as in self-weighted objectives, derive reweighting signals directly from the target policy, which tightly couples the optimization signal to the current model state and can amplify self-reinforcing biases. Fixed references, on the other hand, are mainly used to regularize updates through divergence constraints, and do not adapt to the distribution of verified demonstrations.

In RGFT, the reference policy $\pi_{\text{ref}}$ is intended to represent the data distribution of $\mathcal{D}$, i.e., the conditional distribution over verified solutions observed in the dataset. This suggests a natural training objective: we fit $\pi_{\text{ref}}$ on $\mathcal{D}$ with maximum likelihood, so that it remains well-defined on the support of the demonstrations:

$$\mathcal{L}_{\text{ref}} = -\mathbb{E}_{(x,y)\sim\mathcal{D}} \log \pi_{\text{ref}}(y \mid x). \qquad (4)$$

Importantly, the role of $\pi_{\text{ref}}$ is not to provide a perfect teacher for $\pi_\theta$, but to supply a stable probability scale on verified solutions. The ratio-guided update in (3) uses this scale to upweight correct trajectories that receive relatively low

probability under the current reference, thereby promoting broader coverage than fitting $\pi_{\text{ref}}$ alone. This perspective motivates learning $\pi_{\text{ref}}$ jointly with the target policy, rather than fixing it a priori or tying it implicitly to $\pi_\theta$.

### 3.4. Joint Learning with a Reference Policy

Given a reference policy $\pi_{\text{ref}}$, a natural way to guide the learning of $\pi_\theta$ is through the ratio-based objective (3), which emphasizes correct solutions that are assigned low probability by the reference. We optimize the reference policy using the anchoring objective (4), while updating the target policy using the ratio-guided term (3). This anchoring ensures that $\pi_{\text{ref}}$ remains well-defined on the positive data manifold, without implying that it represents an optimal policy.

A naive joint optimization of (3) and (4) with fixed mixture weights can degenerate under shared initialization. Consider a token position $t$ and denote

$$\pi_{\text{ref}}^{(t)} := \pi_{\text{ref}}(y_t \mid x, y_{<t}), \qquad \pi_\theta^{(t)} := \pi_\theta(y_t \mid x, y_{<t}). \qquad (5)$$

When $\pi_\theta^{(t)} \approx \pi_{\text{ref}}^{(t)}$ (as is typical at the start), the per-token gradients induced by the two terms become nearly aligned because

$$\nabla\mathcal{L}_{\text{ref}} = \nabla\left(-\log \pi_{\text{ref}}^{(t)}\right) = -\frac{1}{\pi_{\text{ref}}^{(t)}}\nabla\pi_{\text{ref}}^{(t)},$$

$$\nabla\mathcal{L}_{\text{ratio}} = \nabla\left(-\frac{\pi_\theta^{(t)}}{\text{sg}(\pi_{\text{ref}}^{(t)})}\right) = -\frac{1}{\text{sg}(\pi_{\text{ref}}^{(t)})}\nabla\pi_\theta^{(t)}. \qquad (6)$$

If the two policies start from identical parameters and are updated with the same optimizer dynamics, this gradient alignment causes $\pi_\theta$ to track $\pi_{\text{ref}}$ closely, effectively collapsing the single-stage procedure into learning a single supervised policy. In turn, the ratio-guided signal becomes less effective at upweighting correct reasoning paths that are underrepresented by the reference, undermining the goal of broadening correct reasoning paths.

### 3.5. Final Objective

To make reference-guided updates effective for *broadening correct reasoning paths*, we must prevent the target policy $\pi_\theta$ from trivially tracking the reference $\pi_{\text{ref}}$ under shared initialization. We therefore introduce a randomized mixture objective that breaks gradient alignment while preserving the intended roles of the two policies. During training, we apply the objective at the token level: for each token position $t$, we sample an independent mixing coefficient $\beta_t \sim \text{Beta}(a, b)$ and optimize

$$\mathcal{L} = -\mathbb{E}_{(x,y)\sim\mathcal{D}}\left[\sum_{t=1}^{|y|}\left(\beta_t \log \pi_{\text{ref}}^{(t)} + (1 - \beta_t)\frac{\pi_\theta^{(t)}}{\text{sg}\left(\pi_{\text{ref}}^{(t)}\right)}\right)\right]. \qquad (7)$$

For intuition, (7) admits a gradient-equivalent reformulation that groups the two terms inside a single fraction:

$$\mathcal{L} = -\mathbb{E}_{(x,y)\sim\mathcal{D}} \left[ \sum_{t=1}^{|y|} \frac{\beta_t\, \pi_{\text{ref}}^{(t)} + (1-\beta_t)\, \pi_\theta^{(t)}}{\text{sg}\left(\pi_{\text{ref}}^{(t)}\right)} \right]. \quad (8)$$

Equation (8) is not a pointwise numerical identity to (7); rather, it is introduced to make the gradient structure explicit. Under the stop-gradient operator, the two formulations induce identical gradients with respect to both $\theta$ and the reference parameters. In particular, the $\beta_t$-weighted term yields the same $\nabla \log \pi_{\text{ref}}^{(t)}$ contribution as in (7), while the $(1-\beta_t)$-weighted term produces the same ratio-guided gradient $\nabla \pi_\theta^{(t)} / \text{sg}(\pi_{\text{ref}}^{(t)})$. As a result, the two objectives induce identical parameter update directions and relative magnitudes under standard first-order gradient-based optimizers (e.g., SGD or Adam), and are therefore equivalent in effect for optimization.

The per-token randomness breaks batch-level gradient colinearity between the two policies. Concretely, over a minibatch $\mathcal{B}$, the reference-policy gradient aggregates terms of the form $\sum_{(x,y)\in\mathcal{B}} \sum_t \beta_t \nabla \log \pi_{\text{ref}}^{(t)}$, while the target-policy gradient aggregates $\sum_{(x,y)\in\mathcal{B}} \sum_t (1-\beta_t) \nabla \pi_\theta^{(t)} / \text{sg}(\pi_{\text{ref}}^{(t)})$. If $\beta_t$ were a fixed constant, these two aggregated gradients can remain nearly proportional when $\pi_\theta \approx \pi_{\text{ref}}$, reinforcing synchronous evolution. By sampling $\beta_t$ independently across tokens (and across examples), the two gradient sums are no longer constrained to be colinear, enabling $\pi_\theta$ and $\pi_{\text{ref}}$ to follow distinct optimization trajectories even under shared initialization.

For additional stability, we optionally include a regularization term that penalizes deviation of the target policy from a base policy $\pi_{\text{base}}$:

$$\mathcal{L}_{\text{KL}} = \lambda\, \mathbb{E}_x \left[ D_{\text{KL}}(\pi_\theta(\cdot \mid x) \,\|\, \pi_{\text{base}}(\cdot \mid x)) \right], \quad (9)$$

where $\pi_{\text{base}}$ denotes the shared initialization of $\pi_\theta$ and $\pi_{\text{ref}}$. This term only constrains the magnitude of policy updates and is orthogonal to the core joint learning mechanism.

The full training objective is given by the sum of (8) and the optional regularization (9).

# 4. Experiments

We evaluate RGFT against standard Supervised Fine-Tuning (SFT) and representative rollout-free reweighting methods, including Dynamic Fine-Tuning (DFT) (Wu et al., 2025) and Anchored Supervised Fine-Tuning (ASFT) (Zhu et al., 2025). Our experiments focus on two questions: (1) whether RGFT consistently outperforms rollout-free baselines across model scales, and (2) whether such gains are achieved by broadening coverage over correct reasoning paths rather

than by excessively sharpening the model distribution, as mentioned in (Yue et al., 2025). Following common practice, we report Pass@1 accuracy averaged over 16 runs and examine Pass@k at large $k$ to assess generalization under sampling. We further include auxiliary analyses of distributional behavior and optimization stability to support these findings.

## 4.1. Experimental Setup

We evaluate RGFT across multiple model families and scales. Specifically, we use Qwen2.5-Math-1.5B, Qwen2.5-Math-7B (Qwen Team, 2024), Qwen3-1.7B, and Qwen3-4B (Qwen Team, 2025) as base models.

We construct the training set following (Wu et al., 2025) by randomly sampling 100,000 examples from the NuminaMath-CoT dataset (LI et al., 2024), which contains diverse mathematical problems with step-by-step chain-of-thought solutions. We report performance on five mathematical reasoning benchmarks: MATH500 (Hendrycks et al., 2021), AIME 2024 (American Institute of Mathematics, 2024), AMC 2023 (Mathematical Association of America, 2023), Minerva Math (Lewkowycz et al., 2022), and OlympiadBench (He et al., 2024).

Our implementation is built upon the Verl framework (Sheng et al., 2024). We train all models for 1 epoch using AdamW (Loshchilov & Hutter, 2017) with a global batch size of 256. The learning rate is set to $2 \times 10^{-5}$ for models with parameters $\geq 4B$ and $5 \times 10^{-5}$ otherwise. Unless otherwise specified, the mixing coefficient is sampled as $\beta \sim \text{Beta}(10, 10)$ at each training step, which introduces controlled stochasticity in the joint objective and improves optimization robustness.

We report Pass@1 accuracy averaged over 16 independent runs with temperature 1.0. To assess generalization under sampling and the coverage over correct reasoning paths, we further report Pass@k for $k \in \{1, 2, 4, 8, 16, 32, 64, 128\}$ on MATH500, Minerva Math, and OlympiadBench, and for $k \in \{256, 512, 1024\}$ on AIME 2024 and AMC 2023.

## 4.2. Main Results

**Consistent gains across model scales.** Table 1 summarizes Pass@1 results across four base models. RGFT consistently improves over standard SFT and representative rollout-free baselines (DFT and ASFT) across all evaluated model families and scales. For example, on Qwen2.5-Math-7B, RGFT achieves 44.65% average Pass@1, improving over the strongest baseline DFT (37.88%) by +6.77%. Similar gains hold from smaller models such as Qwen2.5-Math-1.5B (+1.94% over DFT) to larger models such as Qwen3-4B, indicating that the method scales reliably and does not hinge on a particular capacity regime.

*Table 1.* Main results comparing our method against SFT and reweighting baselines across different model scales. We report the Pass@1 accuracy (%) averaged over 16 runs on five benchmarks. The best results are highlighted in **bold**.

| Base Model | Method | MATH500 | Minerva | Olympiad | AIME 2024 | AMC 2023 | Avg. |
|---|---|---|---|---|---|---|---|
| Qwen2.5-Math-1.5B | Base | 31.66 | 8.51 | 15.88 | 4.16 | 19.38 | 15.92 |
| | SFT | 43.54 | 12.31 | 11.96 | 0.78 | 17.58 | 17.23 |
| | DFT | 64.00 | 23.74 | 26.47 | 5.57 | 38.98 | 31.75 |
| | ASFT | 63.75 | 24.06 | 26.73 | 4.84 | 38.52 | 31.58 |
| | Ours | **67.10** | **25.18** | **27.90** | **7.87** | **40.39** | **33.69** |
| Qwen2.5-Math-7B | Base | 40.09 | 13.74 | 17.04 | 8.75 | 27.66 | 21.46 |
| | SFT | 52.91 | 19.13 | 17.84 | 2.49 | 25.04 | 23.48 |
| | DFT | 68.35 | 34.87 | 33.81 | 8.34 | 44.02 | 37.88 |
| | ASFT | 70.61 | 30.90 | 33.46 | 9.16 | 45.19 | 37.86 |
| | Ours | **75.69** | **37.60** | **37.53** | **16.10** | **56.33** | **44.65** |
| Qwen3-1.7B | Base | 71.25 | 32.43 | 30.85 | 8.02 | 47.15 | 37.94 |
| | SFT | 36.13 | 10.59 | 9.21 | 0.73 | 13.40 | 11.68 |
| | DFT | 54.95 | 19.79 | 21.31 | 2.38 | 31.35 | 21.63 |
| | ASFT | 69.80 | 28.66 | 31.54 | 10.21 | 43.32 | 30.59 |
| | Ours | **74.31** | **33.88** | **35.58** | **15.26** | **51.33** | **42.07** |
| Qwen3-4B | Base | 70.13 | 34.67 | 30.25 | 10.42 | 43.01 | 37.70 |
| | SFT | 54.05 | 27.89 | 19.78 | 1.45 | 22.27 | 25.09 |
| | DFT | 66.68 | 34.98 | 30.31 | 7.93 | 45.74 | 37.13 |
| | ASFT | **83.91** | 44.53 | 44.23 | **24.07** | 63.36 | 52.02 |
| | Ours | 82.19 | **47.64** | **44.34** | 20.73 | **67.30** | **52.44** |

**Stronger performance on challenging benchmarks.** Improvements are particularly pronounced on harder benchmarks such as AIME 2024 and OlympiadBench, which require long-horizon and brittle multi-step reasoning where a single misstep can invalidate an entire solution chain. On AIME 2024 with Qwen2.5-Math-7B, RGFT reaches 16.10% Pass@1, nearly doubling the performance of DFT (8.34%) and substantially outperforming SFT (2.49%). Similarly, on OlympiadBench, RGFT achieves 37.53% compared to 33.81% for DFT and 17.84% for SFT. These results suggest that RGFT is more effective at improving difficult instances under positive-only supervision, where aggressive self-weighting methods can suffer from gradient instability due to extreme importance weights. We further analyze sampling-based generalization (Pass@k) and distributional behavior in the next section to clarify *how* these gains are achieved.

**Avoiding degradation on strong base models.** A notable observation on Qwen3 models is that standard SFT causes severe performance degradation rather than improvement. On Qwen3-1.7B, SFT drops average accuracy from 37.94% to 11.68%, and on Qwen3-4B from 37.70% to 25.09%. This phenomenon aligns with prior findings that fine-tuning on external data can distort pretrained features (Kumar et al., 2022) or lead to catastrophic forgetting of existing capabilities (Luo et al., 2023), particularly when the base model has already been extensively optimized for the target domain. DFT partially mitigates this but still underperforms Base on Qwen3-1.7B (21.63% vs 37.94%). In contrast, RGFT

avoids degradation and consistently improves over Base (37.94% → 42.07% on Qwen3-1.7B; 37.70% → 52.44% on Qwen3-4B), suggesting that reference-anchoring helps preserve pre-existing reasoning capabilities.

### 4.3. Broad Coverage without Excessive Sharpening

We examine whether RGFT's gains come from *broader coverage over correct reasoning paths* under sampling, rather than from aggressively sharpening the model distribution. We use Pass@k scaling as a coverage proxy under sampling, and analyze distributional behavior via token-level probabilities, log-ratio dynamics, and KL divergence to the base model.

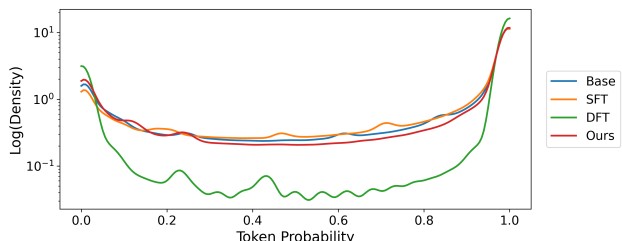

*Figure 1.* Token probability distribution on training data. DFT shows heavy densities at extreme values, indicating over-confident predictions. Our method maintains a distribution similar to base model and SFT, preserving exploration capability.

**Not achieved by sharpening the distribution.** A potential explanation for improved Pass@k is that the fine-tuned policy becomes over-confident (i.e., lower-entropy or

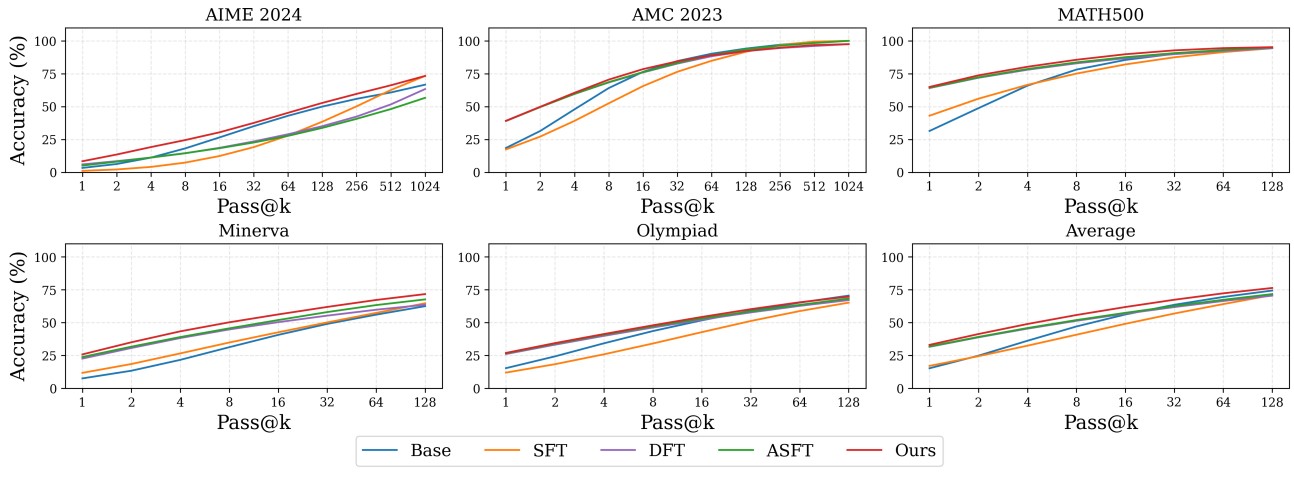

*Figure 2.* Comparison of Pass@k accuracy across different methods on five mathematical reasoning benchmarks. We evaluate five approaches: Base, SFT, DFT, ASFT, and Ours.

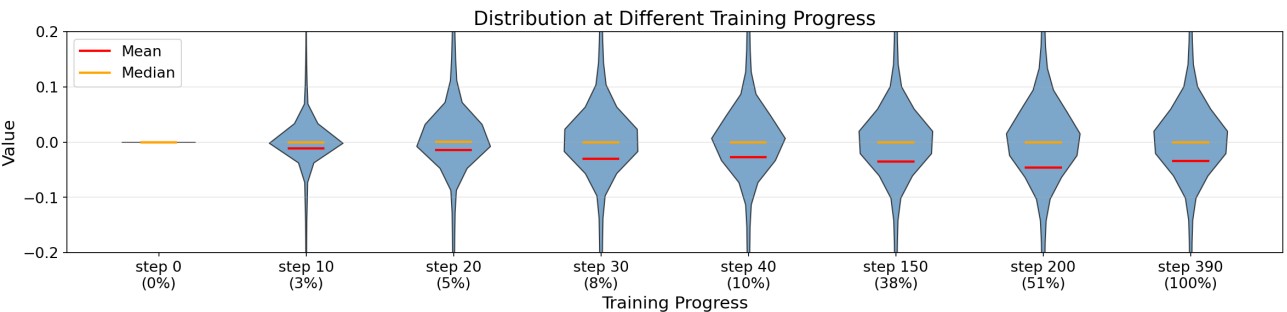

*Figure 3.* Distribution of $\log(\pi_\theta/\pi_{\text{ref}})$ across training steps. The variance increases slowly over training (suggesting increased variability in $\pi_\theta$), while the median remains close to zero and the distribution does not develop heavy tails, consistent with staying close to the reference policy.

sharper token distributions), which can increase Pass@1 but often harms sampling diversity and robustness. Figure 1 visualizes token probability distributions on the training data. DFT exhibits a pronounced U-shape with heavy density near 0 and 1, indicating strongly polarized token probabilities and near-deterministic predictions. In contrast, RGFT remains close to Base and SFT, retaining substantial mass in the mid-probability range. This indicates that RGFT does not obtain gains by aggressively concentrating probability mass, but instead preserves uncertainty at the token level. Taken together, these observations suggest that RGFT improves performance without inducing destructive probability polarization, supporting the view that its Pass@k gains arise from broader coverage rather than from distribution sharpening.

**Broader coverage under sampling (Pass@k).** Figure 2 reports Pass@k scaling across benchmarks. A common phenomenon in RL-style (Yue et al., 2025) or aggressively reweighted objectives is that, while they may improve Pass@1 or small-$k$ performance, their advantage often diminishes or even reverses at larger $k$, where base models

or SFT can recover due to broader and less polarized distributions. In contrast, RGFT consistently outperforms Base, SFT, DFT, and ASFT across the entire range of $k$ values.

This behavior suggests that RGFT achieves a more balanced trade-off between exploitation and coverage. Rather than concentrating probability mass on a narrow set of preferred trajectories—as is typical in sharper or self-weighted objectives—RGFT assigns non-negligible probability to a wider set of correct reasoning paths. As a result, its advantage is maintained from small to large $k$, instead of being overtaken by less specialized baselines at high sampling budgets.

*Table 2.* KL Divergence between fine-tuned models and the base model. Lower values indicate closer alignment to pre-trained knowledge.

| Method | $D_{\text{KL}}(\pi_{\text{base}}\|\pi)$ | $D_{\text{KL}}(\pi\|\pi_{\text{base}})$ |
|--------|------|------|
| SFT | **0.0072** | **0.0074** |
| DFT | 0.4754 | 0.2962 |
| Ours | 0.0107 | 0.0103 |

*Table 3.* Ablation on $\beta$ mixing strategies using Qwen2.5-Math-1.5B. We report Pass@1 (%) averaged over 16 runs on five benchmarks.

| $\beta$ Strategy | MATH500 | Minerva | Olympiad | AIME 2024 | AMC 2023 |
|---|---|---|---|---|---|
| $B(10, 10)$ | **67.10** | 25.18 | **27.90** | **7.87** | **40.39** |
| $B(20, 20)$ | 65.13 | 26.42 | 26.81 | 7.19 | 40.16 |
| Log decay | 63.90 | **26.39** | 26.41 | 7.55 | 38.09 |
| Exp decay | 63.86 | 24.71 | 26.05 | 7.18 | 38.75 |
| Two-stage | 66.56 | 24.65 | 27.73 | 7.30 | 36.68 |
| Fixed 0.5 | 61.96 | 23.60 | 23.30 | 4.84 | 22.50 |

**Stable distributional behavior during training.** We further quantify distributional drift from the base model in Table 2. Compared with DFT, RGFT exhibits substantially smaller KL divergence to $\pi_{\text{base}}$ and stays close to SFT, indicating limited drift from pre-trained knowledge. Figure 3 provides a complementary view by tracking the distribution of $\log(\pi_\theta/\pi_{\text{ref}})$ over training. While the variance increases gradually, the median remains near zero and the distribution does not develop heavy tails, consistent with $\pi_\theta$ staying close to the learned reference scale rather than drifting arbitrarily. Together, these results suggest that RGFT improves sampling-based generalization while maintaining a stable, non-peaked distribution.

### 4.4. Ablation on $\beta$ Mixing Strategies

RGFT relies on a per-token mixing coefficient $\beta_t$ to stochastically interpolate between the reference-log-likelihood term and the ratio-guided target update. To understand the role of this stochastic mixing, we compare different choices of $\beta$ on Qwen2.5-Math-1.5B (Table 3). We consider: (i) *stochastic* $\beta_t$ sampled from symmetric Beta distributions, which injects small, persistent noise into gradient mixing; (ii) *deterministic* schedules (log/exp decay) that gradually shift the mixture over training; (iii) a *two-stage* variant that switches from pure reference fitting to pure ratio guidance; and (iv) a *constant* mixture $\beta \equiv 0.5$, which can make the two objectives overly aligned under shared initialization. Overall, stochastic mixing yields the best and most robust performance, while deterministic or stage-wise schedules are more prone to degradation on harder benchmarks.

Across benchmarks, sampling $\beta_t$ from a symmetric Beta distribution performs best overall, with $\text{Beta}(10, 10)$ giving the strongest average. Deterministic decay schedules and the two-stage switch can reduce performance, especially on harder benchmarks, consistent with the importance of maintaining stochastic mixing throughout training. Using a constant mixture ($\beta \equiv 0.5$) performs worst, supporting the intuition that fixed mixing can lead to overly aligned updates under shared initialization.

## Conclusion

We presented Reference-Guided Fine-Tuning (RGFT), a rollout-free framework for improving mathematical reasoning under positive-only supervision. Unlike existing approaches that rely on fixed or implicit references, RGFT introduces a learnable reference policy that is jointly optimized with the target policy, providing a data-grounded probability scale over correct reasoning paths. A Beta-distributed stochastic mixing scheme breaks gradient alignment between the two policies, enabling them to assume distinct functional roles during training.

Experiments across four model families and five mathematical benchmarks demonstrate that RGFT consistently outperforms standard SFT and recent rollout-free reweighting methods. Crucially, these gains are achieved without the destructive probability polarization observed in aggressive reweighting approaches: RGFT maintains low KL divergence to the base model while improving Pass@k scaling at large $k$, indicating broader coverage over valid reasoning trajectories rather than over-confident sharpening. We believe the core idea of jointly learning a reference policy may generalize to other positive-only settings and provide stronger initialization for subsequent RL stages.

## Impact Statement

This paper presents work whose goal is to advance the field of machine learning. There are many potential societal consequences of our work, none of which we feel must be specifically highlighted here.

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
