# OpenReview forum: "RGFT: Broadening Correct Reasoning Paths via Reference-Guided Fine-Tuning"
_ICML.cc/2026/Conference — Submitted to ICML 2026_

### Official Review · Reviewer_ZubW · 2026-03-08

**Soundness:** 2
**Presentation:** 1
**Significance:** 3
**Originality:** 2
**Overall Recommendation:** 2
**Confidence:** 2

**Summary:**

This is paper introduces a method RGFT which is better than SFT in many ways. It shows better overall performance and also show no degradation unlike SFT on better models. It uses two policy: references policy and target policy. Target policy update using reference policy which reference policy is majority the distribution map of correct (x,y) samples. To prevent collapse in the start of gradient update because both target and reference policy start from the same initialized model, they introduce a beta term.

**Compliance With Llm Reviewing Policy:**

Affirmed.

**Final Justification:**

No rebuttal response provided.

**Key Questions For Authors:**

1. (Check soundness for details): Why not present the paper as a general method which performs better than SFT. It will make the work more impactful.
2. What is the theoretical justification for this method working ?

**Limitations:**

1. No theoretical justification.
2. Over presentation clarity like subparagraphs for related works, citations missing.
3. The paper being presented only for reasoning LLM setting limits the impact of the work. If the method actually works, it should be presented a general approach to post-training.

**Strengths And Weaknesses:**

Soundness: The experimental results are dense with multiple models but it's only for reasoning data. Its seems like the paper is giving a method which is better than SFT and SFT is utilized in many scenarios. The overall paper doesn't make sense to focus only on reasoning tasks. It should have been presented a general method applicable to any scenario like vision tasks, coding tasks and should have been tested on different types architectures.

Presentation: Presentation needs to improve include grammar mistakes overall in the paper. (1) Section 3.2 "D" is undefined. (2) Section 3.1: Citations would have been good especially when talking about previous methods. Related works should have had titles. First paragraph of introduction just mentions things without citing anything. This is not very formal.

Significance: The paper lacks theoretical justification on why the method works. It does provide intuition but to present this as better post training approach, it needs some theoretical justification.

Originality: I am sure about the previous works. I believe similar ideas should have been tested. A more thorough survey of previous methods, not just for LLMs should be presented.

---

### Official Review · Reviewer_WWxv · 2026-03-10

**Soundness:** 2
**Presentation:** 1
**Significance:** 2
**Originality:** 3
**Overall Recommendation:** 2
**Confidence:** 4

**Summary:**

This paper proposes an SFT-like objective that trained only on the correct solutions, and avoid coverage collapse over valid reasoning
paths.  The proposed RGFT introduces a learnable reference policy and a ratio-guided objective to encourage correct reasoning paths with low likelihood. Additionally, mix training objective is proposed to prevent tracking behavior in optimization. Experiments show the effectiveness.

**Compliance With Llm Reviewing Policy:**

Affirmed.

**Final Justification:**

I will keep my score.

**Key Questions For Authors:**

1. The learning objective for policy and reference is not clear, whether we adopt Eq. 7 for both of them or only policy.
2. The transition from Eq. 2 to Eq. 3 is not clear for data distribution.
3. Whether the performance gain came from learned reference distribution, or from mixture final objective, the ablations is lacked.

**Limitations:**

The authors haven't include limitations in their work.

**Strengths And Weaknesses:**

**Strengths:**
1. The authors proposes a learned policy to capture data distribution that avoiding diversity collapse in SFT optimization.
2. Experiments on Math benchmarks show the effectiveness of the method;

**Weaknesses:**
1. The paper lacks validation that whether RGFT truly guarantees better coverage of reasoning paths, the provided pass@k results seems not convincing with minor improvement compared to SFT objective, and even lower in AMC23;
2. It should be validated that whether the learned reference distribution converge to something meaningful?
3. The authors use a learned reference model to capture data distribution through maximaizing likelihood, and why it is better than self-weighting, additionally, training cost should be provided for comparison.
4. The soundness of mixture final objective is not guaranteed, it should be further clarified.
5. The presentation needs to be improved, and there are a lot of unfamiliar statement that hinder understanding, such as "implicit references", "ratio-guided", "probability scale", etc. Further, there is even no supplementary provided.

---

### Official Review · Reviewer_rR89 · 2026-03-12

**Soundness:** 2
**Presentation:** 3
**Significance:** 2
**Originality:** 3
**Overall Recommendation:** 3
**Confidence:** 4

**Summary:**

This paper investigates methods for optimizing policy models in offline scenarios with only positive samples. The authors point out that existing rollout-free fine-tuning methods lack an explicitly defined probabilistic scale for correct solutions, causing the model to shrink to a few dominant correct inference paths. Therefore, the authors propose Reference-Guided Fine-Tuning (RGFT), introducing a reference policy jointly trained with the target policy, treating it as the distribution of correct solutions in the training data, thus encouraging the model to improve on correct but low-probability trajectories. To address potential degradation issues in joint training, a token-level beta distribution random mixture target is designed. Experiments on models of various sizes demonstrate that the method improves model performance and generalization ability. However, a comparison with some important baseline methods is lacking, as is an analysis of the necessity of its design components.

**Compliance With Llm Reviewing Policy:**

Affirmed.

**Key Questions For Authors:**

1. What are the similarities and differences between the proposed method and iw-SFT? It is recommended to include performance comparisons with iw-SFT.
2. There is a lack of analysis of the reference policy. Since the reference policy and the target policy are jointly trained, what is the performance trend of the reference policy during training? Can the authors provide analyses similar to those for the target policy, including the training dynamics and the final performance of the reference policy?
3. Can the authors provide results for the following experimental settings:
   1. Directly perform SFT training for two epochs.
   2. Replace $\pi_\text{ref}$ in the final training objective with a model that has been finetuned on the dataset distribution (e.g., an SFT model), keep its parameters frozen throughout the training process, and only optimize the policy.
   3. Replace $\pi_\text{ref}$ in the final training objective with the initial (untrained) target policy, keep its parameters frozen throughout the training process, and only optimize the policy.
   4. Similar to the replacements in settings (2) and (3), train the model using the objective defined in Equation (3) together with Equation (9).

**Limitations:**

yes

**Strengths And Weaknesses:**

### Strengths
1. The motivation of the paper is clear. It's reasonable and meaningful to encourage broader correct reasoning paths under the supervised fine-tuning setting.
2. The proposed framework that jointly trains a reference policy and a target policy is novel.
3. The authors provide extensive experiments demonstrating that the proposed method improves the generalization ability of the model.


### Weaknesses
1. Although iw-SFT is mentioned in the related work, it is not included as a baseline in the experiments. The proposed method is structurally similar to iw-SFT. Therefore, more explicit experimental comparisons and a clearer analysis of the differences between the two approaches are necessary.
2. The paper needs to further justify the necessity of introducing a learnable reference policy. In fact, the subsequent mixed objective design is introduced to mitigate the degeneration caused by jointly learning the reference policy. If a reference policy is first trained on the positive demonstration data and then kept fixed, would the aforementioned degeneration issues still arise?
3. The advantages brought by joint training are not sufficiently analyzed. It remains unclear why jointly training the reference and target policies would help the target policy broaden its reasoning paths. In particular, it is not clear whether the performance gain truly comes from the joint training mechanism itself, or simply from the regularization effect introduced by additional stochasticity (randomized mixture objective).

---

### Official Review · Reviewer_rJpS · 2026-03-13

**Soundness:** 3
**Presentation:** 2
**Significance:** 3
**Originality:** 3
**Overall Recommendation:** 4
**Confidence:** 2

**Summary:**

This paper introduces RGFT, an offline and rollout-free training framework to improve LLM reasoning from an offline dataset of correct solutions only. The RGFT approach includes a two-part loss function to improve both a learnable reference model and a target policy model. Specifically, the reference policy is progressively fit to the offline dataset, while the target policy is trained to increase the probability of correct tokens in the offline dataset that are unlikely for the reference model. The intuition of this approach is that this objective broadens the coverage of the space of correct solutions, which will lead to better downstream correctness. The authors instantiate their loss by dynamically mixing the reference fit term with a policy ratio term, including a stop grad on the reference policy to broaden the coverage. The authors motivate this sort of mixing through a gradient analysis, showing potential collapse without some dynamic mixing. RGFT is evaluated on a suite of 6 challenging mathematics benchmarks. The authors also include various analyses including to determine what happens to the likelihood of reference tokens during training and ablations on the mixing approach.

**Compliance With Llm Reviewing Policy:**

Affirmed.

**Final Justification:**

The authors did not engage in the rebuttal, so I will keep the same score as my concerns remain.

**Key Questions For Authors:**

1. Did the authors try using their trained model as an initialization for online RL? It seems like this learned broader policy might serve as a good prior, as entropy seems to be preserved while the model has also learned broader reasoning.

2. How did the authors pick the beta distribution as well as the distribution parameters? Was there some validation?

**Limitations:**

yes

**Strengths And Weaknesses:**

### Strengths
1. This paper presents a novel and clearly motivated loss formulation that seems to enable a learnable reference model for the first time, which enables the target policy to learn from correct offline tokens, while broadening reasoning behaviors.

2. The authors demonstrate that their method has strong empirical performance across a suite of 6 math benchmarks and four base models, outperforming prior methods (DFT, SFT, ASFT)  pass@1.

3. The authors also provide evidence that RGFT makes more modest changes to the base policy compared to competitors like DFT, which is attractive in order to maintain abilities and knowledge learned during pretraining.

### Weaknesses
1. There are a few key areas/assumptions that are not well motivated or founded throughout the paper. For instance, it is assumed that having a learnable reference model is preferred compared to a fixed reference policy. However, there is no empirical or theoretical evidence for this in the paper, as SFT, DFT, and ASFT do not all seem to use a reference policy. It seems like the authors should compare to some other baselines like PSFT. Also, the beta distribution is used, but there is no comparison to other types of distributions, nor is there any discussion of why Beta was specifically chosen.

2.  While this RGFT operates in the rollout-free setting, it would be helpful to see how it compares to online methods. For instance, it would help if the authors compared RL methods like GRPO or DAPO, or online distillation methods like on-policy self-distillation. The authors may find it helpful to contextualize the differences in performance with some quantification of compute or cost differences between these two settings.

3. There are a few presentation issues. Figure 2 has all curves scaled between 0 and 100, which makes it difficult to see the difference between methods at higher values of k. Additionally, density is not adequately defined in Figure 1's caption nor in Section 4.3.

---

### Decision · Program_Chairs · 2026-04-30

**Decision:**

Reject

**Comment:**

The paper introduces RGFT, an offline and rollout-free training method to improve LLM reasoning from an offline dataset that only contains the correct solutions. The reviewers raised questions such as the lack of comparison to online methods and the presentation of the paper. The authors did not respond. Therefore, I believe this paper requires major revision before publication.